# Comparing labour induction outcomes using misoprostol and dinoprostone in term pregnancies: A retrospective study at Kiambu Level 5 Hospital between 2018 and 2020

Magoma Mwancha-Kwasa[1], Rashida Admani[2], Margaret Mbuga[2], Mary Maina[2], Jonathan Mwangi[3], Lucy Ng'ang'a[2], Margaret Waweru[2], Sarah Mwangi[2], Patrick Nyaga[1], Davis Kamondo[1], Grace Akech Ochieng[2], Prabhjot Kaur Juttla[4]*, Ryan Nyotu[1], Teresia Njoki Kimani[1], Moses Ndiritu[1]

1 Department of Health, County Government of Kiambu, Kiambu, Kiambu County, Kenya, 2 Kiambu Level Five Hospital, County Government of Kiambu, Kiambu, Kiambu County, Kenya, 3 School of Pharmacy and Health Sciences, United States International University, Nairobi, Kenya, 4 Faculty of Health Sciences, School of Medicine, University of Nairobi, Nairobi, Kenya

* pkjuttla13@gmail.com

## Abstract

### Background

The Maternal and Perinatal Death Surveillance and Response (MPDSR) was introduced in Kenya in 2016 and implemented at Kiambu Level 5 Hospital (KL5H) three years later in 2019. During a routine MPDSR meeting at KL5H, committee members identified a possible link between the off-label use of 200mcg misoprostol tablets divided eight times to achieve the necessary dose for labour induction (25mcg) and maternal deaths. Following this, an administrative decision was made to switch from misoprostol to dinoprostone for the induction of labour in June of 2019. This study aimed to assess the overall impact of MPDSR as well as the effect of replacing misoprostol with dinoprostone on uterine rupture, maternal and neonatal deaths at KL5H.

### Methods

We conducted a retrospective cohort study of women who gave birth at KL5H between January 2018 and December 2020. We defined the pre-intervention period as January 2018—June 2019, and the intervention period as July 2019–December 2020. We randomly selected the records of 411 mothers, 167 from the pre-intervention period and 208 from the intervention period, all of whom were induced. We used Bayes-Poisson Generalised Linear Models to fit the risk of uterine rupture, maternal and perinatal death. 12 semi-structured key person questionnaires was used to describe staff perspectives regarding the switch from misoprostol to dinoprostone. Inductive and deductive data analysis was done to capture the salient emerging themes.

### Results

We reviewed 411 patient records and carried out 12 key informant interviews. Mothers induced with misoprostol (IRR = 3.89; CI = 0.21–71.6) had an increased risk of death while

**Data Availability Statement:** All relevant data are within the manuscript and its Supporting Information files (S1 File).

**Funding:** The author(s) received no specific funding for this work.

**Competing interests:** The authors have declared that no competing interests exist.

mothers were less likely to die if they were induced with dinoprostone (IRR = 0.23; CI = 0.01–7.12) or had uterine rupture (IRR = 0.56; CI = 0.02–18.2). The risk of dying during childbearing increased during Jul 2019–Dec 2020 (IRR = 5.43, CI = 0.68–43.2) when the MPDSR activities were strengthened. Induction of labour (IRR = 1.01; CI = 0.06–17.1) had no effect on the risk of dying from childbirth in our setting. The qualitative results exposed that maternity unit staff preferred dinoprostone to misoprostol as it was thought to be more effective (fewer failed inductions) and safer, regardless of being more expensive compared to misoprostol.

## Conclusion

While the period immediately following the implementation of MPDSR at KL5H was associated with an increased risk of death, the switch to dinoprostone for labour induction was associated with a lower risk of maternal and perinatal death. The use of dinoprostone, however, was linked to an increased risk of uterine rupture, possibly attributed to reduced labour monitoring given that staff held the belief that it is inherently safer than misoprostol. Consequently, even though the changeover was warranted, further investigation is needed to determine the reasons behind the rise in maternal mortalities, even though the MPDSR framework appeared to have been put in place to quell such an increase.

## Background

Induction of labour (IOL) is the iatrogenic process of initiating uterine contractions in pregnant women who are not in labour in an attempt to achieve vaginal delivery [1]. IOL is indicated when the benefits of delivery outweigh the risks of continuing the pregnancy for the mother and/or the foetus [1]. Labour is induced by softening, effacing and dilating the cervix (cervical ripening), and the two major techniques are mechanical interventions or the use of pharmacological agents like prostaglandins (PG) [2].

Misoprostol and Dinoprostone are both synthetic PG analogs that exert effects on $PGE_1$ and $PGE_2$ receptors respectively [3]. Both of these induction agents have been approved by the World Health Organization (WHO) [4]. Low doses (25 mcg every 4 hours) of oral or vaginal misoprostol are used for IOL. Alternatively, it is given in the form of a low dose oral Misoprostol solution, known as "Cytotec solution" [3]. Administered thus, it is thought to have equal safety and efficacy profiles to dinoprostone [5]. The latter is administered intra-vaginally and is the PG agent most widely recommended for ripening of the cervix [6, 7].

Labour induction exceeds 20% of all births in most developed countries [2]. This is different from the case in Africa, where the relatively scant data reveals that rates of labour induction are lower [8]. For instance, in a 2002 Nigerian study, only 3% of women were induced [9]. There is a direct connection between induction of labour and the health-related Sustainable Development Goals because it has the potential to promote maternal well-being and safe delivery [8, 10]. Reasons for induction include: preeclampsia beyond 37 weeks gestation, chorioamnionitis, non-reassuring foetal status, deteriorating maternal condition, post-datism and term prelabour rupture of membranes [8, 11]. However, IOL is contraindicated in previous uterine rupture, cephalo-pelvic disproportion, foetal malpresentation and relatively contraindicated in previous caesarean sections. While inducing labour using pharmacologic interventions is helpful, it also increases the risk of uterine hyperstimulation, caesarean sections, operative vaginal

delivery, chorioamnionitis, cord prolapse, abruption of the placentae and uterine ruptures [11]. It is also not without consequences for the foetus, as frequent uterine activity decreases neonatal brain oxygenation [12, 13].

The cost of IOL using misoprostol is Kenya Shillings (KSh.) 160 (USD 1.00) for a complete dose. In comparison, three tablets of Dinoprostone costs the patient a total of KSh. 3000 (USD 18.75). Misoprostol is a low-cost drug and can be stored at room temperature (unlike dinoprostone)–making it the preferred agent in resource-poor settings [3]. In addition, it is the only drug provided by the Kenya Medical Supplies Agency (KEMSA) for IOL as a 200mcg tablet.

The WHO introduced the Maternal Death Surveillance and Response (MDSR) approach in 2012 as a new method for maternal death review [14]. It built upon MDSR by including perinatal death surveillance, becoming MPDSR. This framework approaches the management of a maternal client as being multidisciplinary and explores system failures that may have led to mortality [14]. Contextual quality improvement solutions are then developed and implemented. In Kenya, two of the WHO policy indicators have been met: first in 2004 by establishing a national policy for notifying maternal deaths and in 2014 when a national committee for MPDSR was established [14]. A third WHO policy was achieved in July of 2019 by establishing a facility-based MPDSR team at Kiambu Level 5 Hospital (KL5H).

During a routine MPDSR meeting at KL5H, it was observed that misoprostol had been used for IOL in a number of maternal mortalities. The committee pinned these maternal mortalities on the splitting of the 200mcg tablet that is only scored once (as supplied by KEMSA) into eight pieces for IOL at KL5H. Following this observation, the use of misoprostol for IOL was suspended and replaced with dinoprostone. This decision was made in June of 2019. To assess the basis of this decision, we retrospectively reviewed the records of mothers who delivered at KL5H during Jan 2018—Dec 2020 to assess both the (1) effect of the implementation of MPDSR and (2) switch to dinoprostone on the outcomes of uterine rupture, maternal and perinatal deaths after induction of labour. Further, we obtained qualitative feedback from the health providers from the maternity unity regarding the two interventions.

## Methods

### Study design

This was a mixed-methods approach combining a retrospective cohort study conducted using data from January 2018 to December 2020 with key informant interviews providing in-depth qualitative dimensions.

### Study site

The study was conducted in Kiambu Level 5 Hospital (KL5H), Longitude 36.83098, Latitude -1.17209, in Kiambu County, Kenya. It is a public health facility located 13 kilometres north of Nairobi, the capital city of Kenya. The catchment population of the hospital is 101,596 with women of reproductive age (15–49 years) constituting a population of 28,305 (27.86%). On average, KL5H maternity labour ward reports around 800 deliveries per month. The paediatric unit has a new born unit and paediatric wards but no Neonatal Intensive Care Unit.

### Local context

Prior to June 2019, the drug used for IOL was misoprostol because it is supplied by the Kenya Medical Supplies Authority (KEMSA) with the cost (KSh. 160, USD 1.00 per dose) being covered by *Linda Mama* (a maternity health cover administered by the national government), and

therefore free to the patients. Misoprostol is stored at room temperature while dinoprostone requires a cold chain, making the former the preferred agent in resource-poor settings [3]. Misoprostol as supplied by KEMSA is a 200mcg tablet, and, as the dose required for induction is 25mcg, the tablet is split to achieve this dosage. After June 2019, dinoprostone was recommended but it was paid for out of pocket by the patients (KSh 3000, USD 18.75 per dose) and is not covered by *Linda Mama*.

### Induction protocols during the period of study

During the study period, the protocol for misoprostol use was vaginal route with 25mcg administered every 4 hours after review with a maximum of 8 doses. The protocol for dinoprostone use was vaginal route with 3 mg administered every 6–8 hours after review with a maximum of 3 doses. Vaginal misoprostol or dinoprostone were only administered by medical officers stationed in the maternity unit. In the case of administering misoprostol solution (Cytotec solution), nurses in the maternity unit were tasked with its preparation, administering it to patients and monitoring their ingestion of it within the specified timeframe.

### Sampling

**Sample size determination.** Data on the labour induction rate for KL5H was unavailable. This study used the figures reported for Kenyatta National Hospital (KNH) of 12.7% [15]. This value is that of the largest Tertiary Referral Centre in our country and which is approximately 18.2 Km away from KL5H. We used Fisher's formula to calculate the sample size, as below:

$$n = Z^2 PQD^2$$

Where 'n' represents the sample size, 'Z' equals 1.96 for a 95% confidence interval, 'P' denotes the proportion of mothers induced for labour using the KNH values, 'Q' stands for the proportion of mothers not requiring induction, and 'D' represents the absolute error (0.05). Using the above formula and proportion of mothers induced, the minimum required sample size is equal to 170 for those who were induced and 170 for those who were not induced.

**Sampling procedure.** The study reviewed records spanning a 36-month period which worked out to 10 randomly picked data records per month for the study period: January 2018 to December 2020.

We conducted 12 key informant interviews utilizing a population-based approach, wherein all eligible staff members who were invited to participate and who provided informed consent were included in the study.

*Inclusion criteria.* The inclusion criteria consisted of the data records pertaining to women who underwent induction within the designated timeframe.

For the qualitative aspect, eligible participants included heads of the maternity unit, registered clinical officers with specialization in reproductive health and nurses stationed in the maternity unit, each possessing a minimum of five years' experience within the hospital's maternity ward.

Furthermore, these individuals were required to have rotated throughout the study period, spanning from January 2018 to December 2020.

*Exclusion criteria.* The exclusion criteria entailed data records or patient files that were deemed unsuitable for data abstraction due to conditions such as tearing or incompleteness.

Regarding the qualitative segment, exclusion criteria encompassed the registered clinical officers specialized in reproductive health and nurses within the maternity unit who did not attend rotations scheduled during the study period. Additionally, medical officers were

excluded due to their rotational assignments across various departments every 3–6 months, rendering their continuity within the maternity unit inconsistent.

## Data collection

Secondary data collection was done using patient files from the maternity and new born units using a pre-designed e-tool. This data was collected retrospectively from the 21st of May to the 27th May 2021. Only the variables pertinent for the study were collected. The dataset containing deidentified information related to the study can be located in the supplementary file labeled as S1 File. The following variables were collected: age, occupation, education level, marital status, parity, gestation, induction agent, mode of delivery, maternal outcome, newborn outcome, uterine rupture, post-partum haemorrhage and study period.

A self-administered key informant questionnaire was used to capture qualitative data on the impact of MPDSR (S2 File) and the change of labour induction agent. These interviews were conducted from the 21st of May to the 27th May 2021.

## Data analysis

Cross-tabulation was used to summarise patient characteristics against induction of labour. The Pearson's Chi-square of independence and Fisher's exact test were used to test the relationship between patient characteristics and induction of labour. Three outcomes of interest were identified: maternal mortality, uterine rupture, and neonatal mortality. We ran 11 explanatory variables against each outcome and tabulated unadjusted and adjusted point estimates and their 95% confidence intervals. The explanatory variables were chosen based on their relationship to the exposure and outcomes, as well as their clinical relevance. We calculated the relative risk of each outcome by fitting a Bayesian Poisson Generalised linear model to the data. The Bayesian model accounts for predictor multicollinearity by selecting and clustering predictors at the same time. The analyses were carried out using the R version 4.1.2 (Copyright© 2021 The R Foundation for Statistical Computing).

The qualitative data underwent both inductive and deductive data analysis. Three reviewers (MK, RN and PKJ) read through the data and individually developed the initial codes through indexing and charting. The emerging themes were documented, discrepancies resolved and final themes agreed upon by consensus.

## Ethical considerations

Ethics approval was obtained from the Baraton University Ethics Review Committee, approval number: UEAB/REC/09/05/2021. Because retrospective secondary data collection was done, there was no direct contact with patients, and no consent was therefore sought from the patients, and at no point during the study was there any direct interaction or contact with any of the patients, nor was their identifying or contact information gathered or stored.

Before participation, all respondents of the key informant questionnaire provided written, signed, and informed consent. Although the tools were self-administered, study personnel ensured that health workers were consented individually before completing the forms, with each participant signing consent forms prior to their engagement. No identifying data was collected from participants before, during, or after the completion of the self-administered key informant questionnaires. Furthermore, no incentives were offered for questionnaire participation, which was voluntary and had no bearing on participants' job or employment status.

## Results

### Participant characteristics

Within the study 36-month study period, 518 mothers underwent labour induction out of 10,236; giving an induction rate of 5.1% at the facility for the period between January 2018 and December 2020.

The study had 411 patients, the majority of whom (96%, n = 394) were between the ages of 18 and 40, and unemployed (62%, n = 234). The highest educational attainment for the participants was 28% (n = 103), 50% (n = 188) and 22% (n = 82) for primary, secondary and tertiary education respectively. Nearly all (95%, n = 318) were married, with 49% (n = 201) being nulliparous. As shown in Table 1, the majority of the patients' characteristics were distributed equally between those who were induced and those who were not. Of the study participants analysed, 24% (n = 96) had a Caesarean section, 6.3% (n = 25) had postpartum haemorrhage, 4.5% (n = 18) had a ruptured uterus, and 3.4% (n = 14) died. Twenty-six (6.5%) of the babies born died at birth or within seven days of birth.

During the study period, misoprosotol was used to induce 21.2% of the women. Out of these, a maternal death occurred in 4.5% (n = 5). 7.3% of the inductions failed. Just 1 (0.9%) of the women who had misoprostol inductions experienced uterine rupture. During the study period, 18.3% of the women underwent induction using dinoprosotone. Out of these, 1 (1.1%) led to a maternal death, while 16.9% (n = 16) of the induction attempts failed. Uterine rupture occurred in 7.4% (n = 7) of the women who had dinoprostone induction.

### Risk of maternal death during childbirth

Mothers induced with misoprostol (IRR = 3.89; CI = 0.21–71.6), who delivered by Cesarean Section (IRR = 3.27; CI = 0.45–24.0), who had pre-eclampsia/eclampsia (IRR = 11.3; CI = 0.63–17.9), mothers with 1–2 (IRR = 1.28; CI = 0.18–9.34) or more than 2 babies (IRR = 8.44; CI = 0.92–77.0) had an increased risk of death.

Mothers were less likely to die if they were induced with dinoprostone (IRR = 0.23; CI = 0.01–7.12), had uterine rupture (IRR = 0.56; CI = 0.02–18.2), were under the age of 18 (IRR = 0.83; CI = 0.01–54.8) or over the age of 40 (IRR = 0.42; CI = 0.01–12.4), had received tertiary education (IRR = 0.39; CI = 0.01–10.4), were single (IRR = 0.58; CI = 0.02–20.9), or presented for delivery at a gestation age of 42 weeks and more (IRR = 0.7; CI = 0.02–30.6).

The risk of dying during childbearing increased during Jul 2019–Dec 2020 (IRR = 5.43, CI = 0.68–43.2) when the MPDSR activities were strengthened. Induction of labour (IRR = 1.01; CI = 0.06–17.1) had no effect on the risk of dying from childbirth (Table 2).

### Risk of perinatal death

There was an increased adjusted risk of perinatal death for babies whose mothers suffered uterine rupture (IRR = 1.66; CI = 0.22, 12.3), or were induced with misoprostol (IRR = 2.96; CI = 0.27, 32.4). Babies born between Jan 2019 and Dec 2020 when MPDSR was revitalised (IRR = 1.81; CI = 0.46, 7.14) had a higher risk of dying perinatally. Conversely, there was a decreased adjusted relative risk of death among babies born to mothers who were induced with dinoprostone (IRR = 0.75; CI = 0.07, 8.43) or 18 years old (IRR = 0.75; CI = 0.01, 38.3). Unlike for maternal deaths at childbirth, induction of labour resulted in a higher risk (IRR = 1.50, CI = 0.14, 16.1) of perinatal death (Table 3).

### Risk of uterine rupture

Mothers who were induced with dinoprostone (IRR = 1.97; CI = 0.21–18.9) or delivered before 37 completed weeks (IRR = 2.85; CI = 0.82–9.89) had an increased adjusted risk for uterine

**Table 1. Characteristics of study participants stratified according to induction of labour.**

| Characteristic | n | Overall, n = 411[1] | Labour Induced | | p-value[2] |
| --- | --- | --- | --- | --- | --- |
| | | | No, n = 203[1] | Yes, n = 208[1] | |
| **Age (years)** | 411 | | | | 0.6 |
| **<18** | | 9 (2.2%) | 6 (3.0%) | 3 (1.4%) | |
| **18–40** | | 394 (96%) | 193 (95%) | 201 (97%) | |
| **>40** | | 8 (1.9%) | 4 (2.0%) | 4 (1.9%) | |
| **Occupation** | 379 | | | | 0.28 |
| **Employed** | | 144 (38%) | 64 (35%) | 80 (41%) | |
| **Unemployed** | | 235 (62%) | 118 (65%) | 117 (59%) | |
| **Education level** | 373 | | | | 0.78 |
| **Primary** | | 103 (28%) | 53 (29%) | 50 (27%) | |
| **Secondary** | | 188 (50%) | 94 (51%) | 94 (50%) | |
| **Tertiary** | | 82 (22%) | 38 (21%) | 44 (23%) | |
| **Marital status** | 335 | | | | 0.54 |
| **Married** | | 318 (95%) | 155 (96%) | 163 (94%) | |
| **Single** | | 17 (5.1%) | 7 (4.3%) | 10 (5.8%) | |
| **Parity** | 410 | | | | 0.94 |
| **0** | | 201 (49%) | 98 (48%) | 103 (50%) | |
| **1–2** | | 162 (40%) | 82 (40%) | 80 (39%) | |
| **>2** | | 47 (11%) | 23 (11%) | 24 (12%) | |
| **Gestation(weeks)** | 387 | | | | <0.001 |
| **28–37** | | 74 (19%) | 37 (20%) | 37 (18%) | |
| **38–41** | | 278 (72%) | 144 (77%) | 134 (67%) | |
| **≥42** | | 35 (9.0%) | 5 (2.7%) | 30 (15%) | |
| **Induction agent** | 410 | | | | <0.001 |
| **None** | | 208 (51%) | 203 (100%) | 5 (2.4%) | |
| **Dinoprostone** | | 94 (23%) | 0 (0%) | 94 (45%) | |
| **Misoprostol** | | 108 (26%) | 0 (0%) | 108 (52%) | |
| **Mode of delivery** | 404 | | | | 0.05 |
| **SVD** | | 308 (76%) | 160 (80%) | 148 (72%) | |
| **CS** | | 96 (24%) | 39 (20%) | 57 (28%) | |
| **Maternal outcome** | 411 | | | | 0.56 |
| **Alive** | | 397 (97%) | 195 (96%) | 202 (97%) | |
| **Died** | | 14 (3.4%) | 8 (3.9%) | 6 (2.9%) | |
| **Newborn Outcome** | 404 | | | | 0.02 |
| **Alive** | | 378 (94%) | 194 (97%) | 184 (91%) | |
| **Died** | | 26 (6.4%) | 7 (3.5%) | 19 (9.4%) | |
| **Uterine rupture** | 398 | 18 (4.5%) | 9 (4.6%) | 9 (4.5%) | 0.95 |
| **Postpartum haemorrhage** | 398 | 25 (6.3%) | 12 (6.1%) | 13 (6.5%) | 0.86 |

[1] n (%)

rupture. The risk of uterine rupture increased during Jul 2019 –Dec 2020 (IRR = 16.1; CI = 0.92–284) when MPDSR activities were also strengthened. Mothers with post-partum haemorrhage (IRR = 0.41; CI = 0.88–2.05) or pre-eclampsia/eclampsia (IRR = 0.26; CI = 0.01–5.02) had a lower risk of uterine rupture (Table 4).

## Qualitative results

**Work experience and respondent characteristics.** There were twelve (12) key informants. The average duration of the respondents who had worked in maternity units was 4.5

**Table 2. Risk of death for women delivering at Kiambu Level 5 Hospital: January 2018–December 2020.**

| Characteristic | n | Univariate | | | Multivariable | | |
|---|---|---|---|---|---|---|---|
| | | IRR[1] | 95% CI[1] | p-value | IRR[1] | 95% CI[1] | p-value |
| **Labour induction** | 411 | | | | | | |
| No | | Ref | Ref | | Ref | Ref | |
| Yes | | 0.75 | 0.24–2.11 | 0.6 | 1.01 | 0.06, 17.1 | >0.9 |
| **Age (years)** | 415 | | | | | | |
| 18–40 | | Ref | Ref | | Ref | Ref | |
| <18 | | 0.49 | 0.02–12.9 | 0.7 | 0.83 | 0.01, 57.1 | >0.9 |
| >40 | | 0.51 | 0.02–14.2 | 0.7 | 0.41 | 0.01, 11.7 | 0.6 |
| **Occupation** | 382 | | | | | | |
| Employed | | Ref | Ref | | Ref | Ref | |
| Unemployed | | 2.15 | 0.61–16.4 | 0.3 | 2.37 | 0.34, 16.3 | 0.4 |
| **Education level** | 377 | | | | | | |
| Primary | | Ref | Ref | | Ref | Ref | |
| Secondary | | 0.75 | 0.22–2.61 | 0.7 | 4.25 | 0.46, 39.6 | 0.2 |
| Tertiary | | 0.42 | 0.07–2.51 | 0.3 | 0.37 | 0.01, 9.88 | 0.6 |
| **Marital status** | 338 | | | | | | |
| Married | | Ref | Ref | | Ref | Ref | |
| Single | | 1.51 | 0.09–8.86 | 0.7 | 0.64 | 0.02, 25.1 | 0.8 |
| **Parity** | 414 | | | | | | |
| 0 | | Ref | Ref | | Ref | Ref | |
| 1–2 | | 3.57 | 0.99–12.9 | 0.05 | 1.64 | 0.25, 11.0 | 0.6 |
| >2 | | 5.74 | 1.34–24.6 | 0.02 | 10.4 | 1.09, 98.6 | 0.04 |
| **Gestation(weeks)** | 391 | | | | | | |
| 38–41 | | Ref | Ref | | Ref | Ref | |
| 28–37 | | 3.4 | 0.91–12.7 | 0.07 | 2.3 | 0.36, 14.7 | 0.4 |
| ≥42 | | 0.38 | 0.02–8.52 | 0.5 | 0.69 | 0.02, 30.2 | 0.8 |
| **Induction agent** | 414 | | | | | | |
| None | | Ref | Ref | | Ref | Ref | |
| Dinoprostone | | 0.36 | 0.07–1.94 | 0.2 | 0.21 | 0.01, 6.63 | 0.4 |
| Misoprostol | | 1.2 | 0.41–3.51 | 0.7 | 4.34 | 0.24, 79.6 | 0.3 |
| **Mode of delivery** | 408 | | | | | | |
| SVD | | Ref | Ref | | Ref | Ref | |
| CS | | 2.09 | 0.46–10.8 | 0.3 | 4.24 | 0.65, 27.6 | 0.13 |
| **Pre-eclampsia** | 415 | 4.62 | 1.20–17.4 | 0.02 | 11.2 | 0.61, 203 | 0.1 |
| **Uterine rapture** | 402 | | | | | | |
| No | | Ref | Ref | | Ref | Ref | |
| Yes | | 0.51 | 0.02–14.3 | 0.7 | 0.56 | 0.02, 18.0 | 0.7 |
| **Study period** | 414 | | | | | | |
| Jan 2018—Jun 2019 | | Ref | Ref | | Ref | Ref | |
| Jul 2019—Dec 2020 | | 1.1 | 0.38–3.62 | 0.9 | 5.78 | 0.75, 44.6 | 0.09 |

[1]IRR = Incidence Rate Ratio, CI = Confidence Interval

years and the average duration working specifically at the KL5H maternity unit was found to be 2 years. Five (5) of the 12 respondents have never worked at another maternity unit other than KL5H.

**Table 3. Risk of death for babies delivered at Kiambu Level 5 Hospital: January 2018–December 2020.**

| Characteristic | n | Univariate | | | Multivariable | | |
|---|---|---|---|---|---|---|---|
| | | IRR[1] | 95% CI[1] | p-value | IRR[1] | 95% CI[1] | p-value |
| Labour induction | 404 | | | | | | |
| No | | Ref | Ref | | Ref | Ref | |
| Yes | | 2.55 | 1.18–6.89 | 0.03 | 1.55 | 0.14–16.7 | 0.7 |
| Age (years) | 408 | | | | | | |
| 18–40 | | Ref | Ref | | Ref | Ref | |
| <18 | | 0.36 | 0.02–7.73 | 0.5 | 0.74 | 0.01–38.1 | 0.9 |
| >40 | | 1.87 | 0.28–12.5 | 0.5 | 2 | 0.25–15.8 | 0.5 |
| Occupation | 375 | | | | | | |
| Employed | | Ref | Ref | | Ref | Ref | |
| Unemployed | | 1.11 | 0.49–2.77 | 0.8 | 1.68 | 0.51–5.59 | 0.4 |
| Education level | 370 | | | | | | |
| Primary | | Ref | Ref | | Ref | Ref | |
| Secondary | | 1.09 | 0.40–3.57 | 0.9 | 2.86 | 0.75–10.9 | 0.12 |
| Tertiary | | 1.27 | 0.37–4.74 | 0.7 | 1.61 | 0.28–9.13 | 0.6 |
| Marital status | 332 | | | | | | |
| Married | | Ref | Ref | | Ref | Ref | |
| Single | | 1.7 | 0.29–6.36 | 0.5 | 2.56 | 0.31–21.1 | 0.4 |
| Parity | 407 | | | | | | |
| 0 | | Ref | Ref | | Ref | Ref | |
| 1–2 | | 3.01 | 1.25–7.25 | 0.01 | 2.15 | 0.60–7.74 | 0.2 |
| >2 | | 2.5 | 0.76–8.30 | 0.13 | 7.21 | 1.47–35.4 | 0.01 |
| Gestation(weeks) | 384 | | | | | | |
| 38–41 | | Ref | Ref | | Ref | Ref | |
| 28–37 | | 4.32 | 1.94–11.7 | <0.001 | 2.59 | 0.76–8.84 | 0.13 |
| ≥42 | | 1.56 | 0.26–6.87 | 0.5 | 4.74 | 0.82–27.3 | 0.08 |
| Induction agent | 407 | | | | | | |
| None | | Ref | Ref | | Ref | Ref | |
| Dinoprostone | | 1.74 | 0.61–5.78 | 0.3 | 0.75 | 0.07–8.33 | 0.8 |
| Misoprostol | | 3.28 | 1.47–9.62 | 0.01 | 3.09 | 0.28–33.9 | 0.4 |
| Mode of delivery | 401 | | | | | | |
| SVD | | Ref | Ref | | Ref | Ref | |
| CS | | 0.76 | 0.21–1.99 | 0.6 | 1.77 | 0.45–6.89 | 0.4 |
| Pre-eclampsia | 408 | 3.79 | 1.22–11.0 | 0.01 | 7.65 | 1.27–46.1 | 0.03 |
| Uterine rupture | 395 | | | | | | |
| No | | Ref | Ref | | Ref | Ref | |
| Yes | | 1.13 | 0.07–5.63 | 0.9 | 1.66 | 0.22–12.2 | 0.6 |
| Study period | 407 | | | | | | |
| Jan 2018—Jun 2019 | | Ref | Ref | | Ref | Ref | |
| Jul 2019—Dec 2020 | | 0.85 | 0.39–1.87 | 0.7 | 1.9 | 0.49–7.38 | 0.3 |

[1]IRR = Incidence Rate Ratio, CI = Confidence Interval

**Theme 1: Cost.** There was clear consensus that Misoprostol is the cheaper drug. Dinoprostone was noted to be "*very expensive*" with the effect of this being that there was prolonged hospital stay for clients as they awaited funds to buy the drug.

Those patients who were unable to afford dinoprostone were reported to have the cost waived or alternative misoprostol formulations used.

**Table 4. Risk of uterine rupture among mothers delivering at Kiambu Level 5 Hospital.**

| Characteristic | n | Univariate | | | Multivariable | | |
|---|---|---|---|---|---|---|---|
| | | IRR[1] | 95% CI[1] | p-value | IRR[1] | 95% CI[1] | p-value |
| **Labour induction** | 398 | | | | | | |
| No | | Ref | Ref | | Ref | Ref | |
| Yes | | 0.97 | 0.38–2.49 | >0.9 | 0.89 | 0.10–8.23 | >0.9 |
| **Age (years)** | 402 | | | | | | |
| 18–40 | | Ref | Ref | | Ref | Ref | |
| <18 | | 0.44 | 0.02–10.5 | 0.6 | 0.59 | 0.02–20.4 | 0.8 |
| >40 | | 2.24 | 0.32–15.8 | 0.4 | 2.79 | 0.31–25.4 | 0.4 |
| **Occupation** | 372 | | | | | | |
| Employed | | Ref | Ref | | Ref | Ref | |
| Unemployed | | 2 | 0.77–7.56 | 0.2 | 1.67 | 0.50–5.56 | 0.4 |
| **Education level** | 369 | | | | | | |
| Primary | | Ref | Ref | | Ref | Ref | |
| Secondary | | 1.32 | 0.46–5.03 | 0.6 | 1.54 | 0.43–5.44 | 0.5 |
| Tertiary | | 1.19 | 0.29–5.27 | 0.8 | 1.45 | 0.35–5.99 | 0.6 |
| **Marital status** | 328 | | | | | | |
| Married | | Ref | Ref | | Ref | Ref | |
| Single | | 3.43 | 0.89–11.9 | 0.05 | 2.69 | 0.71–10.2 | 0.14 |
| **Parity** | 401 | | | | | | |
| 0 | | Ref | Ref | | Ref | Ref | |
| 1–2 | | 1.58 | 0.61–4.62 | 0.3 | 1.21 | 0.42–3.46 | 0.7 |
| >2 | | 1.15 | 0.18–5.01 | 0.8 | 0.58 | 0.08–4.30 | 0.6 |
| **Gestation(weeks)** | 385 | | | | | | |
| 38–41 | | Ref | Ref | | Ref | Ref | |
| 28–37 | | 1.42 | 0.47–4.27 | 0.5 | 2.38 | 0.72–7.83 | 0.2 |
| > = 42 | | 0.22 | 0.01–3.87 | 0.3 | 0.21 | 0.01–3.91 | 0.3 |
| **Induction agent** | 401 | | | | | | |
| None | | Ref | Ref | | Ref | Ref | |
| Dinoprostone | | 1.7 | 0.61–4.65 | 0.3 | 2.24 | 0.23–21.4 | 0.5 |
| Misoprostol | | 0.46 | 0.06–1.62 | 0.3 | 0.67 | 0.06–8.01 | 0.8 |
| **Pre-eclampsia** | 402 | 0.3 | 0.02–5.66 | 0.4 | 0.28 | 0.01–5.62 | 0.4 |

[1]IRR = Incidence Rate Ratio, CI = Confidence Interval

"*The hospital social worker intervenes to ensure the drug is either waived or how we can get the drug for induction*" (**Respondent_1**)

"*Substituted with hoffman's solution*" (**Respondent_10**)

**Theme 2: Effectiveness.** Misoprostol was described as having a lower rate of failed induction than dinoprostone. During the time of misoprostol, there were "*reduced c/s operations done*". Misoprostol has "*fast action times*" and when using it there were "*fewer doses needed*". However, it was noted that use of Misoprostol had adverse "*side effects*" such as uterine rupture, precipitate labour, hyperstimulation of the uterus and postpartum haemorrhage.

Of note is that ruptured uterus and precipitate labour were the commonest side effects of misoprostol listed by those with more than 5 years' experience working in maternity units.

75% of the staff and 100% of those with more than 5 years' experience in the department preferred Dinoprostone over Misoprostol despite the higher cost of acquisition and storage.

75% of respondents with more than eight years' experience said they would not be comfortable with a decision to return to using cytotec as the drug of choice for inducing labour if the study's findings showed no correlation between the use of the medication and poor maternal outcomes. The respondents stated:

"*cytotec to be phased out in induction of labour*" (**Respondent_4**)

"*I hope and wish cytotec will be eliminated in induction of labour*" (**Respondent_5**)

**Theme 3: Ease of administration.** The respondents felt that dinoprostone was easier to administer as it came in the right dose. Some felt that for misoprostol to be used properly, "*every intern to be trained on how to divide the tablets if its 25 mcg let the dose be right*". A further comment was that "*cytotec should come in the right dose (25mcg)*". A challenge with dinoprostone as reported was that unlike misoprostol, it required a cold chain for storage and transport.

## Discussion

There is often a disconnect between medical practice in the context of a clinical trial where conditions are well controlled, and the pragmatic care that is in fact delivered [16]. While most research seeks to ascertain the clinical effectiveness of a specific therapy, this study sought to describe the actual "real world" use of labour induction agents. McGlynn *et al.* found a significant difference between research findings on effective therapies and what is actually practised in clinical settings [17].

Being single, having a job and university education were protective against the risk of maternal death. This is partially in agreement with a global survey conducted by WHO on Maternal and Perinatal Health in 373 health centres [18]. However, in the latter study, being single was found to be associated with maternal death. This difference may be due to single and highly educated women earning more than their married counterparts [19]. This associated increase in wealth may have attributed to quality obstetric care and being well-informed [18, 20].

Multiparity was significantly associated with a 115-increased risk of maternal death. This is akin to a prospective study by Dior *et al.* which showed a positive association between parity and all-cause mortality in mothers [21]. Although this association has been accepted, the exact mechanism is unknown. It has been posited to result from a myriad of factors such as: physical demands of repeated pregnancies, anemia, low maternal weight gain, advancing maternal age, lack of pregnancy spacing, lipodystrophy and dietary factors [22].

We showed that women who delivered via CS were four-times more likely to result in mortality. A case-controlled study from France and a population-based cohort from California align with these findings [23, 24]. This outcome may be precipitated by anaesthetic complications, puerperal infections and venous thromboembolism. Furthermore, a diagnosis of preeclampsia increases the risk for maternal death, which mirrors a review by Duley L [25].

## Effect of the MPDSR committee and it's decision

In our study, the labour induction rate in the facility over a 36-month period was 4.4%. This was higher than a study assessing 83,437 deliveries in seven African countries [8], where the rate of induction was 4.4%. There was a 1:1 ratio of the women who were induced with misoprostol and dinoprostone. The proportions of these characteristics are in agreement with an RCT that compared oral misoprostol and vaginal dinoprostone [26].

We found that inducing labour using misoprostol was associated with a greater risk of maternal mortality compared to dinoprostone. This is contrary to two studies comparing the agents which found no differences in maternal deaths [27, 28]. This incongruence may be due to the lack of a standardised dose of misoprostol as used in the KL5H setting contributing to dose related adverse effects in the current study.

However, we found that the risk of maternal deaths increased significantly after the implementation of MPDSR. Our findings contradict the posited effects of implementing MPDSR [29]. This is similar to a study conducted in Ethiopia in which the desired effects were not achieved [30]. This may be due to low-quality reviews and poor continuity between MPDSR audits and quality improvement at the facility level [31]. However, it may also be due to other factor(s) contributing to maternal death, such as the high number of referrals to the hospital, and not necessarily a result of the implementation of the MPDSR committee.

The risk of maternal and newborn deaths increased significantly after the implementation of MPDSR. This is in contrast to the posited effects of implementing an MPDSR audit to avoid future maternal and perinatal deaths and improve outcomes [29]. This effect may be due to the aforementioned reasons for maternal deaths, but may be also due to the lack of a NICU providing specialised care in KL5H.

## Effect on uterine rupture

Patients with uterine rupture were less likely to die. This is unlike the findings in a study conducted in Northwest Ethiopia where uterine rupture was associated with maternal death [32]. Post-dated pregnancy was also associated with a 30% reduction in maternal death. This is in agreement with a study conducted in Cologne where post-dated pregnancies had comparable outcomes to expectantly managed women [33]. In the current study, both effects may have resulted from prompt identification and management of obstetric complications coupled to increased surveillance of already high-risk pregnancies and/or induced women, which is in line with the WHO standards [4].

Protective factors against the risk of uterine rupture included being below 18 years, while being above 40 years carried a higher likelihood of uterine rupture. This association between advancing maternal age and uterine rupture has been previously shown in a 12-year retrospective study [34]. The effect is pinned on the likelihood of higher parity in older women. Other characteristics that increased the risk of uterine rupture included being unemployed, obtaining higher education and being single. The characteristics of women managed for uterine rupture in a retrospective analysis conducted in Ado-Ekiti contrast these findings [35]. This may be due to a lack of access to obstetric care, less compliance to medical instruction and lack of empowerment among these women in the present study.

Inducing labour was found to be protective against uterine rupture. This is in contrast to a prospective observational study that showed women who underwent trial of labour had increased odds of uterine rupture [36]. However, this study was conducted in women who had previously delivered via CS. Multiparity and term-pregnancies were found to have reduced associated risks with uterine rupture. This is in contrast to a study on the predictors of uterine rupture conducted in Mali and Senegal [37]. The incompatibility between the studies may be due to better monitoring and surveillance structures that are already in place for these high-risk groups in the present study preventing deleterious complications.

Dinoprostone was associated with 2.24-fold increased risk of uterine rupture. This contradicts a meta-analysis conducted on the prevalence of uterine rupture after IOL [38]. In addition, after the implementation of MPDSR committee, the risk of uterine rupture increased. Lack of surveillance and regular monitoring after induction due to staffing shortages

combined with the assumption that dinoprostone is generally safer for IOL may have precipitated this observation.

### Effect on perinatal death

Advanced maternal age, unemployment, education and singlehood were associated with approximately two-fold greater risks of newborn deaths. These results are mostly in line with a study conducted in Nigeria [39], however in the current study advancing educational status carried a higher relative risk of newborn death. The associated increase in maternal age and newborn death is probably due to a corresponding increase in maternal morbidities [40].

IOL increased the risk of perinatal death in our setting which is contrary to a multicentre trial that concluded term-labour induction does affect perinatal outcomes [41, 42]. Furthermore, misoprostol increased the risk of perinatal mortality and a study by Morris *et al*. contradicts this finding [43]. It may be due to increased foetal distress which has been reported with the use of misoprostol [44].

Evidence from the current study showed an association between multiparity and perinatal death. This link between high parity and increased perinatal mortality has been documented and corroborates the findings of the current study [22]. In addition, late presentation and a higher likelihood of comorbidities in multiparous women may have contributed to this finding. In the present study, advanced gestation was associated with newborn deaths. This is corroborated with a meta-analysis that found increased risks of neonatal death with prolonged gestation [45]. Asphyxia due to advanced gestation and a larger foetus may have contributed to this finding.

IOL using dinoprostone was found to reduce the risk of perinatal mortality. This is contrary to an RCT comparing misoprostol and dinoprostone where perinatal outcomes were similar between both arms of the study [46]. However, misoprostol was shown to increase the outcome of non-reassuring foetal status due to tachysystole [46]. This effect was associated with precipitate labour and hence the safety associated with dinoprostone may be due to preventing this effect.

### Attitude towards the contextual use of misoprostol

In terms of cost and effectiveness, misoprostol was the preferred agent for IOL. This is in agreement with a study comparing the cost and efficacy of the two agents [47]. Furthermore, women who cannot afford dinoprostone, Cytotec solution is given but this may elicit dose-related adverse effects without proper administration. This highlights a policy gap to be addressed in terms of making dinoprostone more affordable and available given its protective effects against maternal deaths.

The nurses relayed that dinoprostone was easier to administer because it came in the right dosage. To our knowledge, this particular advantage has not been reflected in literature which normally compares the use of 25mcg misoprostol with dinoprostone [3, 5, 48]. In our context, dinoprostone is easier to administer as it comes in the right dose while the 200mcg misoprostol tablet requires to be split manually or made into solutions for use.

### Limitations

The limitations of the current study are that the sampled files were pseudo-randomized and by virtue of being retrospective lacked a standardised tool to collate the required data. However, the study did manage to elucidate the effects of the interventions put in place in a real-world setting which buttresses the application of its findings.

### Recommendations

We suggest reevaluating the current protocols in light of our inquiry of the practical use of Misoprostol in our particular setting, which is representative of practices seen not just in our region but also in many other parts of Kenya and Africa. Although there are established recommendations, our results highlight the particular nuances of our background and local setting, which influence Misoprostol use in ways that deviate from normal protocols.

Given the objectives of our research, we support policy discussions that take into account the implementation of protocols using dinoprostone, exploring avenues to procure the 25 mg dose of Misoprostol, and fostering a broader acknowledgment that current practices, although ingrained, may not align with optimal safety standards. Government subsidies should also shift towards making dinoprostone more affordable to women in our setting.

Furthermore, a clear algorithm taking into account these two common labour induction practices should be implemented in line with the standards set out by the WHO for performing labour induction [4]. The following criteria must be met in all instances of labour induction: there must be access to facilities for evaluating the health of both the mother and the foetus; women who are administered oxytocin, misoprostol, or other prostaglandins should never be left unsupervised; and whenever feasible, induction of labour should take place in a setting equipped to perform caesarean sections.

Lastly, collectively moving beyond the status quo of "*what we have always done*" is essential in our context, as is critically reevaluating our method of administering misoprostol. This recommendation is meant to take centre stage throughout our conversation and act as a strong recommendation for the Ministry of Health (MoH) to take into account in their continuous endeavours to improve healthcare practice at the county levels and nationwide. In our pursuit of ensuring the well-being of mothers, it is imperative that we uphold proper practices without compromise to ensure the best possible maternal outcomes.

### Conclusion

The facility MPDSR committee's decision to transition from misoprostol to dinoprostone was justified regarding the risk of maternal mortality. Dinoprostone, however, was not a perfect solution because of the increased risk of uterine rupture, but this may be due to the general notion that it is safer and requires less monitoring. This calls for the increased surveillance of women induced using dinoprostone to prevent deleterious outcomes and policy changes to reduce the cost of the drug. Overall, the risk of maternal death was higher in the period after the interventions indicating that misoprostol was not the only cause of mortality. Therefore, there is a need to look deeper into the causes of maternal mortality.

### Supporting information

**S1 File. Underlying data for this study.** This Excel sheet contains the deidentified mothers and babies' data underlying this study.
(XLSX)

**S2 File. The qualitative data collected regarding the MPDSR committee.**
(PDF)

### Acknowledgments

This endeavour was made possible by the contributions of numerous individuals and public and private institutions. We are grateful to the Medical Superintendent of Kiambu County

Referral Hospital, the Department of Health Records, and the labour ward, maternity and new born unit for their time, resources, and records, and for answering our numerous clarification questions. We acknowledge The County Government of Kiambu and the County Health Management Team for their support and input into this work. This study would not have been possible without the assistance of the Transforming Health Systems for Universal Care (THS_UC) Project, which supported MPDSR activities in the County of Kiambu and this study's research. Finally, we are grateful to the patients of Kiambu County Referral Hospital for allowing us to learn from their experiences in order to enhance the services provided by the Department of Health.

## Author Contributions

**Conceptualization:** Magoma Mwancha-Kwasa, Rashida Admani, Margaret Mbuga, Mary Maina, Jonathan Mwangi, Lucy Ng'ang'a, Margaret Waweru, Sarah Mwangi, Patrick Nyaga, Davis Kamondo, Grace Akech Ochieng, Ryan Nyotu, Teresia Njoki Kimani, Moses Ndiritu.

**Data curation:** Magoma Mwancha-Kwasa, Rashida Admani, Margaret Mbuga, Jonathan Mwangi, Lucy Ng'ang'a, Prabhjot Kaur Juttla, Ryan Nyotu, Moses Ndiritu.

**Formal analysis:** Rashida Admani, Prabhjot Kaur Juttla, Teresia Njoki Kimani, Moses Ndiritu.

**Funding acquisition:** Magoma Mwancha-Kwasa, Margaret Mbuga.

**Investigation:** Mary Maina, Jonathan Mwangi, Lucy Ng'ang'a, Margaret Waweru, Sarah Mwangi, Patrick Nyaga, Davis Kamondo, Grace Akech Ochieng, Ryan Nyotu, Teresia Njoki Kimani, Moses Ndiritu.

**Methodology:** Magoma Mwancha-Kwasa, Mary Maina, Jonathan Mwangi, Lucy Ng'ang'a, Margaret Waweru, Sarah Mwangi, Patrick Nyaga, Davis Kamondo, Ryan Nyotu, Teresia Njoki Kimani, Moses Ndiritu.

**Project administration:** Magoma Mwancha-Kwasa, Margaret Mbuga, Mary Maina, Margaret Waweru, Davis Kamondo, Moses Ndiritu.

**Resources:** Magoma Mwancha-Kwasa, Rashida Admani, Mary Maina, Margaret Waweru, Sarah Mwangi, Patrick Nyaga, Davis Kamondo, Grace Akech Ochieng, Moses Ndiritu.

**Software:** Teresia Njoki Kimani, Moses Ndiritu.

**Supervision:** Magoma Mwancha-Kwasa, Rashida Admani, Moses Ndiritu.

**Validation:** Rashida Admani, Prabhjot Kaur Juttla, Moses Ndiritu.

**Writing – original draft:** Magoma Mwancha-Kwasa, Prabhjot Kaur Juttla, Moses Ndiritu.

**Writing – review & editing:** Magoma Mwancha-Kwasa, Rashida Admani, Prabhjot Kaur Juttla, Moses Ndiritu.

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
