## [Decision Letter · Decision Letter 0]

23 Apr 2024

PONE-D-24-01559Misoprostol versus Dinoprostone: safety when used for induction in a low-resource settingPLOS ONE

Dear Dr. Juttla,

Thank you for submitting your manuscript to PLOS ONE. After careful consideration, we feel that it has merit but does not fully meet PLOS ONE’s publication criteria as it currently stands. Therefore, we invite you to submit a revised version of the manuscript that addresses the points raised during the review process.

**ACADEMIC EDITOR: **Title1 Reformat the title using PICOS form (who exactly are the population, Details of intervention (route), outcome studies and type of the study.Methods1. Trial registration details2. Full details about sample size calculation (power , outcome ,etc,,,,3. More details about inclusion and exclusion details4. Details about the protocol and management during induction of labor, other oxytocics, analgesics, amniotomy , fetal monitoring, maternal follow up, indications of CS5. Define primary and secondary outcomes6. Other outcomes as fetal distress, mternal satisfaction and duration of 1st, 2nd and 3rd stage

We look forward to receiving your revised manuscript.

Kind regards,

Ahmed Mohamed Maged, MD

Academic Editor

PLOS ONE

Journal Requirements:

Reviewers' comments:

Reviewer's Responses to Questions

**Comments to the Author**

1. Is the manuscript technically sound, and do the data support the conclusions?

Reviewer #1: Yes

Reviewer #2: Yes

2. Has the statistical analysis been performed appropriately and rigorously? 

Reviewer #1: Yes

Reviewer #2: Yes

3. Have the authors made all data underlying the findings in their manuscript fully available?

Reviewer #1: Yes

Reviewer #2: No

4. Is the manuscript presented in an intelligible fashion and written in standard English?

Reviewer #1: Yes

Reviewer #2: Yes

5. Review Comments to the Author

Reviewer #1: Dear author

Its an interesting issue identified and written intelligently but i have the following queries

1. If uterine rupture was not associated with misoprostol then what was the reason of high mortality in the group?

2. The usually observed finding which is supported by many studies is the high risk of uterine rupture with misoprosto rather than dinoprostone, why were the results so grossly different from the available literature?

3. Who was resonsible for inducing the patient?

Reviewer #2: Thanks, for letting me review this interesting article.

below I advised to make this comments that help in introducing this article.

Introduction

1-please provide the advantages of Dinoprostone over misoprostol in induction of labor.

misoprostol was cheaper than Dinoprostone. is Dinoprostone available in your country and how much if it and the patients or hospital afford it ?

Method and results was written very good.

discussion

please could you provide some previous articles that discussed the novelty of the hypnosis.

6. PLOS authors have the option to publish the peer review history of their article (what does this mean?). If published, this will include your full peer review and any attached files.

Reviewer #1: **Yes: **SAIDA ABRAR

Reviewer #2: No

---

## [Author Response · Author response to Decision Letter 0]

14 May 2024

PONE-D-24-01559 

PLOS ONE

Dear Dr. Ahmed Mohamed Maged, MD,

We sincerely appreciate the opportunity to revise the above manuscript in light of your valuable feedback. Below, you will find our responses to the comments provided by the editors and both reviewers in bright blue, bold font.

We acknowledge with gratitude the chance to refine our work and endeavor to ensure that our manuscript aligns with the standards set forth by your esteemed journal.

We believe that the insightful observations made by yourself and the reviewers have significantly enriched our manuscript, and we are truly thankful for them.

We eagerly anticipate your response.

Kind regards, 

Prabhjot Kaur Juttla. 

 

ACADEMIC EDITOR:

Title

1 Reformat the title using PICOS form (who exactly are the population, Details of intervention (route), outcome studies and type of the study. 

Author’s response: Thank you for this comment. We have amended the title to be as below. 

“A comparison of outcomes after the induction of labour in term pregnancies using misoprostol and dinoprostone in a low-resource setting between January 2018 and December 2020”

Methods

 Trial registration details. 

Author’s response: This study was not registered as it did not meet the criteria for a trial. During a routine meeting convened by the MPDSR committee at our hospital (Kiambu Level 5), a common factor identified among mortalities was the use of off-label misoprostol to formulate Hoffman’s solution for induction of labour. Improper dosage and administration, often resulting from the need to cut the tablet into pieces and administer it in water, were significant concerns highlighted by the MPDSR committee due to the fact that the misoprostol dose supplied by the government supplier (KEMSA) came in 200mcg tablets that were only scored ONCE for labour induction, as opposed to the 25mcg dose required. This necessitated the tablet to be cut into EIGHT pieces, and the main concern here was that the incorrect dose would end up being administered to the patient. 

Consequently, the MPDSR committee made the decision to introduce refrigerated dinoprostone for labor induction at the facility, a product not provided by the government drug supplier KEMSA. Mothers were required to fund this change out of pocket, with the implementation commencing in 2019.

Our approach involved reviewing data spanning exactly 1.5 years before and after the implementation of this change (January 2018 – December 2020), a pragmatic, real-life investigation into this change. We sought to determine the validity of the decision made by the MPDSR committee and assess whether misoprostol really did pose a greater risks to mothers administered thus. Ultimately, our aim is to utilize this data to advocate for improvements in the supply of appropriate dosage versions by the government drug supplier or the inclusion of dinoprostone in their supplies to alleviate the cost burden to mothers in our facilities given that it is much more expensive. 

Please see line 100 – 109 of the Introduction as below: 

“ In a routine MPDSR meeting at KL5H, it was observed that misoprostol had been used for IOL in a number of maternal mortalities. The committee pinned these maternal mortalities on the splitting of the 200mcg tablet (as supplied by KEMSA) into eight pieces for IOL at KL5H. Following this observation, the use of misoprostol for IOL was suspended and replaced with dinoprostone. This decision was made in June of 2019. To assess the basis of this decision, we retrospectively reviewed the records of mothers who delivered at KL5H during Jan 2018 - Dec 2020 to assess both the (1) effect of the implementation of MPDSR and (2) switch to dinoprostone on the outcomes of uterine rupture, maternal and perinatal deaths after induction of labour. Further, we obtained qualitative feedback from the health providers on the two interventions.”

Line 113 – 116, as below in the Methods section. 

“Study design

This was a mixed methods approach combining a retrospective cohort study conducted using data from January 2018 to December 2020 with key informant interviews providing in-depth qualitative dimensions.”

See line 126 – 135 of the Methods section:

“Local context

Prior to June 2019, the drug used for IOL was misoprostol because it is supplied by the Kenya Medical Supplies Authority (KEMSA) with the cost (KSh. 160, USD 1.00 per dose) being covered by Linda Mama (a maternity health cover administered by the national government), and therefore free to the patients. Misoprostol is stored at room temperature while dinoprostone requires a cold chain, making the former the preferred agent in resource-poor settings [3]. Misoprostol as supplied by KEMSA is a 200mcg tablet, and, as the dose required for induction is 25mcg, the tablet is split to achieve this dosage. After June 2019, dinoprostone was recommended but it was paid for out of pocket by the patients (KSh 3000, USD 18.75 per dose).”

Line 349 – 355 of the Discussion:

“Discussion 

There is often a disconnect between medical practice in the context of a clinical trial where conditions are well controlled, and the pragmatic care that is in fact delivered [16]. While most research seeks to ascertain the clinical effectiveness of a specific therapy, this study sought to describe the actual “real world” use of labour induction agents. McGlynn et al. found a significant difference between research findings on effective therapies and what is actually practised in clinical settings [17].”

 Full details about sample size calculation (power , outcome ,etc,,,, 

Author’s response: Thank you for this comment and apologies for the omission. We have included a statement for the same in our manuscript under “Sample Size Calculation” in the Methods section, line 144 – 157, as below:

“Sample size determination

Data on the labour induction rate for KL5H was unavailable. This study used the figures reported for Kenyatta National Hospital (KNH) of 12.7% [15]. This value is that of the largest Tertiary Referral Centre in our country and which is approximately 18.2 Km away from KL5H. We used Fisher’s formula to calculate the sample size, as below:

n=Z^2 PQD^2

Where 'n' represents the sample size, 'Z' equals 1.96 for a 95% confidence interval, 'P' denotes the proportion of mothers induced for labour using the KNH values, 'Q' stands for the proportion of mothers not requiring induction, and 'D' represents the absolute error or precision (0.05). 

Using the above formula and proportion of mothers induced, the minimum required sample size is equal to 170 for those who were induced and 170 for those who were not induced.”

 More details about inclusion and exclusion details 

Author’s response: Given that this was a retrospective case-control study, it involved abstraction from medical files in the medical records department as detailed below in the manuscript under Sampling Procedure, line 159 – 182:

“Sampling procedure

The study reviewed records spanning a 36-month period which worked out to 10 randomly picked data records per month for the study period: January 2018 to December 2020. 

We conducted key informant interviews utilizing a population-based approach, wherein all eligible staff members were invited to participate and provided informed consent prior to inclusion in the study.

Inclusion Criteria:

The inclusion criteria consisted of the data records pertaining to women who underwent induction within the designated timeframe. 

For the qualitative aspect, eligible participants included heads of the maternity unit, registered clinical officers with specialization in reproductive health and nurses stationed in the maternity unit, each possessing a minimum of five years' experience within the hospital's maternity ward. 

Furthermore, these individuals were required to have rotated throughout the study period, spanning from January 2018 to December 2020.

Exclusion Criteria:

The exclusion criteria entailed data records or patient files that were deemed unsuitable for data abstraction due to conditions such as tearing or incompleteness. 

Regarding the qualitative segment, exclusion criteria encompassed the registered clinical officers specialized in reproductive health and nurses within the maternity unit who did not attend rotations scheduled during the study period. Additionally, medical officers were excluded due to their rotational assignments across various departments every 3-6 months, rendering their continuity within the maternity unit inconsistent.

 Details about the protocol and management during induction of labor, other oxytocics, analgesics, amniotomy , fetal monitoring, maternal follow up, indications of CS. 

Author’s response: In our retrospective study, we exclusively analyzed patient records. Multiple data points, as detailed in the attached S1 File, were abstracted. Subsequent analysis revealed that the variables, such as those requested, including protocol details and management aspects during induction of labor, other oxytocics, analgesics, amniotomy, fetal monitoring, maternal follow-up, and indications of Cesarean section, were indeed collected and modeled. However, they did not yield significant results. Hence, we opted to focus on secondary outcomes as the most appropriate method for presenting our findings.

This has been described in our data analysis section, as below, between line 201 – 212:

“Data analysis

Cross-tabulation was used to summarise patient characteristics against induction of labour. The Pearson's Chi-square of independence and Fisher's exact test were used to test the relationship between patient characteristics and induction of labour. Three outcomes of interest were identified: maternal mortality, uterine rupture, and neonatal mortality. We ran 11 explanatory variables against each outcome and tabulated unadjusted and adjusted point estimates and their 95% confidence intervals. The explanatory variables were chosen based on their relationship to the exposure and outcomes, as well as their clinical relevance. We calculated the relative risk of each outcome by fitting a Bayesian Poisson Generalised linear model to the data. The Bayesian model accounts for predictor multicollinearity by selecting and clustering predictors at the same time. The analyses were carried out using the R version 4.1.2 (Copyright© 2021 The R Foundation for Statistical Computing).”

 Define primary and secondary outcomes. 

Author’s response: Our primary outcomes aimed to investigate the impact of the MPDSR committee, and to ascertain (if it was significant), the role of the different variables in the secondary outcomes, such as: the role of the induction of labour, amniotomy, induction drug and the indication for C-S etc. After this primary analysis, we then looked at our secondary outcomes, which is what we have presented in our paper: uterine rupture, maternal and perinatal mortality. 

 Other outcomes as fetal distress, mternal satisfaction and duration of 1st, 2nd and 3rd stage. 

Author’s response: Thank you for your acknowledgment. Indeed, these factors can significantly impact the outcome. While all these data points were accessible in the medical files and also collected for data analysis, they did not demonstrate a substantial influence on the outcome in our study. 

 

RESPONSE TO REVIEWERS

Reviewer #1

Author’s response: Thank you for these resources, we have formatted our manuscript according to your style templates. Much appreciated. 

 Please provide additional details regarding participant consent. In the ethics statement in the Methods and online submission information, please ensure that you have specified what type you obtained (for instance, written or verbal, and if verbal, how it was documented and witnessed). If your study included minors, state whether you obtained consent from parents or guardians. If the need for consent was waived by the ethics committee, please include this information. 

Author’s response: Thank you for this. 

We have added details to our ethics consideration section of our manuscript, as below, in line 221 – 237 of our manuscript, as below:

“Ethical considerations

Ethics approval was obtained from the Baraton University Ethics Review Committee, approval number: UEAB/REC/09/05/2021. Because retrospective secondary data collection was done, there was no direct contact with patients, and no consent was therefore sought from the patients, and at no point during the study was there any direct interaction or contact with any of the patients, nor was their identifying or contact information gathered or stored. 

Before participation, all respondents of the key informant questionnaire provided written, signed, and informed consent. Although the tools were self-administered, study personnel ensured that health workers were consented individually before completing the forms, with each participant signing consent forms prior to their engagement. No identifying data was collected from participants before, during, or after the completion of the self-administered key informant questionnaires. Furthermore, no incentives were offered for questionnaire participation, which was voluntary and had no bearing on participants' job or employment status.”

 If uterine rupture was not associated with misoprostol then what was the reason of high mortality in the group? 

Author’s response: Initially, we anticipated being able to elucidate the outcomes based on our analysis. However, misoprostol did not exhibit any association with our secondary outcomes. Consequently, while the decision to switch to dinoprostone was ultimately justified as far as maternal mortality risk was concerned, due to misoprostol's heightened risk of death, we remain uncertain about its specific contribution to mortality. We state the same in our conclusion, line 497 – 507, as below:

“Conclusion

The facility MPDSR committee's decision to transition from misoprostol to dinoprostone was justified regarding the risk of maternal mortality.. Dinoprostone, however, was not a perfect solution because of the increased risk of uterine rupture, but this may be due to the general notion that it is safer and requires less monitoring. This calls for the increased surveillance of women induced using dinoprostone to prevent deleterious outcomes and policy changes to reduce the cost of the drug. Overall, the risk of maternal death was higher in the period after the interventions indicating that misoprostol was not the only cause of mortality. Therefore, there is a need to look deeper into the causes of maternal mortality.”

2. The usually observed finding which is supported by many studies is the high risk of uterine rupture with misoprosto rather than dinoprostone, why were the results so grossly different from the available literature? 

Author’s response: We also noted and discussed this observation in our manuscript's discussion section. Within our setting, we hypothesize that this phenomenon is attributed to LESS VIGILANT labour management practices. Specifically, when patients undergo induction with Misoprostol, medical staff tend to monitor labour more closely and maintain better partograph records, leading to enhanced observation. Conversely, when patients are induced with dinoprostone, healthcare workers perceive the drug as "safer," potentially leading to less vigilant monitoring. 

Please see below our paragraph in the discussion section, line 414 - 419

“Dinoprostone was associated with 2.24-fold increased risk of uterine rupture. This contradicts a meta-analysis conducted on the prevalence of uterine rupture after IOL [37]. In addition, after the implementation of MPDSR committee, the risk of uterine rupture increased. Lack of surveillance and regular monitoring after induction due to sta

---

## [Editor Report · Decision Letter 1]

16 May 2024

A comparison of outcomes after the induction of labour in term pregnancies using misoprostol and dinoprostone in a low-resource setting between January 2018 and December 2020

PONE-D-24-01559R1

Dear Dr. Juttla,

We’re pleased to inform you that your manuscript has been judged scientifically suitable for publication and will be formally accepted for publication once it meets all outstanding technical requirements.

Kind regards,

Ahmed Mohamed Maged, MD

Academic Editor

PLOS ONE
---

## [Editor Report · Acceptance letter]

20 May 2024

PONE-D-24-01559R1 

PLOS ONE

Dear Dr. Juttla, 

I'm pleased to inform you that your manuscript has been deemed suitable for publication in PLOS ONE. Congratulations! Your manuscript is now being handed over to our production team.

Kind regards, 

on behalf of

Professor Ahmed Mohamed Maged 

Academic Editor

PLOS ONE